Robust model predictive control for polytopic uncertain systems via a high-rate network with the FlexRay protocol

Wang Jianhua
Fan Fuqiang
Yu Yanye
Du Shuxin shxdu@zjhu.edu.cn
Guo Xiaorui
Huzhou Key Laboratory of Intelligent Sensing and Optimal Control for Industrial Systems, School of Engineering, Huzhou University , Huzhou, Zhejiang , China
See Chan Hwang
Electronic publication date: 2025 Jan 6
Publication date: 2025
Volume: 11
Electronic Location ID: e2580
Received 2024 May 24; Accepted 2024 Nov 13
Copyright: © 2025 Wang et al.
Copyright year: 2025
Copyright holder: Wang et al.
License: This is an open access article distributed under the terms of the Creative Commons Attribution License, which permits unrestricted use, distribution, reproduction and adaptation in any medium and for any purpose provided that it is properly attributed. For attribution, the original author(s), title, publication source (PeerJ Computer Science) and either DOI or URL of the article must be cited.
License URL: https://creativecommons.org/licenses/by/4.0/

Keywords: FlexRay protocol, High-rate network, OFRMPC, Polytopic uncertain systems, Token-dependent Lyapunov-like

Funding: Natural Science Foundation of Zhejiang LQ22F030011 Natural Science Foundation of Huzhou 2022YZ35 Public Welfare Technology Application and Research Project of Zhejiang LGG21E070001 Huzhou Key Laboratory of Intelligent Sensing and Optimal Control for Industrial Systems 2022-17 This work was supported by the Natural Science Foundation of Zhejiang (LQ22F030011), Natural Science Foundation of Huzhou (2022YZ35), Public Welfare Technology Application and Research Project of Zhejiang (LGG21E070001) and Huzhou Key Laboratory of Intelligent Sensing and Optimal Control for Industrial Systems (grant 2022-17). The funders had no role in study design, data collection and analysis, decision to publish, or preparation of the manuscript.

==============================
In this article, the robust model predictive control (RMPC) problem is investigated for a class of polytopic uncertain systems over high-rate networks whose signal exchanges are scheduled by the FlexRay protocol (FRP). During signal measurement, a high-rate network is applied to broadcast the data from the sensors to the controller efficiently. The FRP including the characteristics of event-triggered mechanism and the time-triggered mechanism is embedded into the high-rate network to regulate the data transmission in a circular period which can improve the flexibility of data transmission. With the aid of the Round-Robin and Try-Once-Discard protocols, a new expression of the measurement model is formulated by the use of certain data holding strategies. Subsequently, taking both high-rate networks and FRP into account, sufficient conditions are obtained by solving a time-varying terminal constraint set of an auxiliary optimization problem. In addition, an algorithm including both off-line and on-line parts is provided to find a sub-optimal solution. Lastly, two numerical simulations are carried out to substantiate the validity of the proposed RMPC strategy which is based on FRP and a high-rate network.

Introduction

In the past decades, model predictive control (MPC) (Kothare, Balakrishnan & Morari, 1996; Arroyo et al., 2022; Ding, 2011; Li et al., 2015), has been considered one of the most advanced control theories because of its powerful ability to handle constraints and widely applied in various fields, such as petrochemical, article industries (Zou, Chen & Li, 2010; Huang, Li & Xi, 2012). It should be pointed out that the parameter-uncertainty is inevitable, hence, robust model predictive control (RMPC) has received widespread attention since it has the capacity to cope with parameter-uncertainty from measurement error, noise, and exogenous disturbances (Su, Tan & Lee, 2013; Deng, Shu & Chen, 2022; Schoeffauer & Wunder, 2021). For a few literatures, the research of RMPC algorithms are based on the assumption that the states are measurable (Zeilinger, Morari & Jones, 2014; Xue et al., 2010), which is on the contrary of the practice systems. Therefore, output feedback has attracted more and more attention from scientists and gets a rich body of results. In Wang, Song & Wei (2019), Hassaan et al. (2021), Efimov, de Souza & Raïssi (2021), output feedback is considered on RMPC through some synthesis approaches. In Ding et al. (2008), the stability of out-put feedback in the framework of RMPC is guaranteed when taking both bounded disturbance and polytopic uncertainty into account. In Efimov, de Souza & Raïssi (2021), in order to solve the problem of discrete-time, constrained linear-varying form disturbances and bounded state, an output feedback MPC is presented. Notice that some research results are based on dynamic output feedback whose performance could be well ensured compared with static output feedback. However, it will increase the computation burden and is not conducive to online algorithms due to its complexity. Hence, we propose a RMPC strategy based on the static output feedback for the polytopic uncertainty system in this article.

With the increasing development of communication technology, network connections are ubiquitous in our lives because of their advantages in high-reliability and easy installation (Zhang, Lam & Xia, 2014; Kamal & Chowdhury, 2022; Du et al., 2022; Wang, Ding & Wang, 2022; Viadero-Monasterio et al., 2022). Such systems are different from the traditional point-to-point connection approach called network control systems (NCSs) which connect the components by network. NCSs have a set of merits compared with the traditional systems. However, some shortages still exist, such as communication delay and data collisions which are caused by the limited bandwidth. For the sake of reducing data collision, a few communication protocols are provided to utilize in NCSs, which include Try Once Discard Protocol (TODP) (Wang et al., 2021; Zou, Wang & Gao, 2016), Round-Robin Protocol (RRP) (Xu et al., 2022; Zhu et al., 2018; Wang et al., 2021; Sheng et al., 2018) and Stochastic Communication Protocol (SCP) (Song et al., 2018; Yuan, Guo & Liu, 2020). The transmission order of the shared communication channel is regulated by resorting to the communication protocol so as to alleviate the communication burden. The underlying idea of communication scheduling protocol is based on the principle that only one communication node has the right to access the network at each time instant.

On the other hand, the transmission rate is very important in system performance. According to different transmission media, communication protocols, there are different transmission rates, the higher transmission rate has better performance. At the same time, the sampling rates also affect the performance of the system. Generally speaking, the sampling rates of certain sensors are much lower than the transmission rates of a communication channel, this type of network is known as a high-rate network. It is obvious that over-sampling is inevitable, thereby resulting in redundant signal transmissions, which increase the burden on limited bandwidth. Consequently, it makes sense to consider communication scheduling protocols in high-rate networks (e.g., Song, Yu & Zhang, 2009; Tabbara & Nesic, 2008; Shen et al., 2020). Nevertheless, most results utilizing the time-triggered and the hybrid communication protocol are overlooked, which constitutes our main purpose to investigate the FRP.

It is worth mentioning that most of the above communication protocols are time-triggering (RRP, SCP) or event-triggering (TODP). FRP is a hybrid that combines the features of both event-triggering and time-triggering. More specifically, the time-triggered transmission that is triggered by time-critical periodic is implemented in static segment and the event-triggered transmission that is decided by the rule based-priority scheduling is carried out in dynamic segment (Tang et al., 2018). Currently, the research on FRP-based control/estimation with NCSs as context has made some achievements. In Liu et al. (2021), Wang, Nešić & Postoyan (2017, 2015), the system performance can be guaranteed when the controller/estimation is equipped with FRP. Notice that most investigations of above take continuous-time as the premise, moreover there is a lack of research in the direction of controllers with discrete-time over high-rate networks. In this article, encouraged by Liu et al. (2021) and Pop et al. (2008), a similar framework of FRP that the static segment and dynamic segment employs RRP and TODP, respectively, is presented. According to the characteristics of pre-transmission data, it can realize the priority transmission of emergency data. The rest data can be transmitted flexibly according to the principle of competition. It should be pointed out that the controller performance will be influenced due to the FRP changing the transmission rule, which attracted us to design an output feedback robust model predictive control (OFRMPC) over a high-rate network under FRP to cope with the side-effect.

In light of above discussions, to design the controller over high-rate network under FRP, some challenges should be addressed: 1. How to unify the static segment and dynamic segment transmission rules.

2. How to fully consider the influence of FRP in high-rate communication channels for the sake of reducing the side effects of the underlying scheduling.

3. How to ensure the performance for controller under FRP in the high-rate network.

4. How to guarantee both the stability of the system and the recursive feasibility of the designed algorithm.

Based on the above challenges, the contributions of this article can be highlighted below: 1. A FRP model for a high rate communication network with respect to MPC is developed.

2. Compared with the traditional RMPC, the RMPC based on FRP and a high rate network designed in this article can solve the problem of limited bandwidth in networked control systems, and has the characteristics of strong anti-interference ability and rolling optimization of the traditional RMPC.

3. The RMPC designed in this article simultaneously considers elements such as polyhedral uncertainty, high-rate networks, and input-output constraints.

4. The on-line and off-line recursive optimization algorithm is given, and a series of stable control sequences are obtained.

The issue of robust model predictive control based on FRP and a high-rate network is studied in this article. By utilizing the scheduling of the FRP, the problem of limited bandwidth is resolved and the performance of the system can be enhanced with the help of a high-rate network. Moreover, due to the involvement of FRP that may cause side-effects, an output-feedback model predictive controller is designed by addressing some auxiliary problems and the stability of system can be ensured by the obtained controller gains. Table 1 is an illustration of the common notations in this article.

Table 1 A symbol table.

Symbol	Denotation	
Rn	The n-dimensional Euclidean space	
Rn×m	The set of all n×m real matrices	
I	The identity matrix	
P>0	The symmetric and positive definite matrix	
||x||	The Euclidean norma vector of a x and ||x||2=xTx	
||⋅||F	The Frobenius norm of Rn space	
|A|	The absolute value of A	
0	The zero matrix	
NT	The transpose of matrix N	
N−1	The inverse of matrix N	
∗	The symmetric part of the symmetric matrix	
diag {⋅⋅⋅}	The block-diagonal matrix	
δ(⋅)	The delta function and δ(M) = 1 when M=0 otherwise δ(M) = 0	
[⋅]w	The wth element of a vector	
Γ	The given convex bounded polyhedral domain	
u¯,x¯	The known scalar	
φ	The robust positive invariant	

Problem statement and preliminaries

A switched system with a high-rate network and polytopic uncertainties under the FRP is proposed in this section. Next, a static output feedback RMPC problem is presented and some preparations for analyzing the performance are illustrated.

System models

The system with polytopic uncertainties is given as follows:

(1) {x(tk+1)=A(tk)x(tk)+B(tk)u(tk)y(tk)=C(tk)x(tk)

where u(tk)∈Rnu, y(tk)∈Rny and x(tk)∈Rnx represent the control input, measure output and state, respectively. A(tk), B(tk) and C(tk) are certain matrices which belong to a convex hull given by

(2) Γ=Δ{Π|Π=∑l=1LϑlΠ(l),∑l=1Lϑl=1,0≤ϑl≤1},

where Π:=(A(tk),B(tk),C(tk))∈Γ, Π(l) are previously known and defined as Π(l):=(A(l),B(l),C(l)),l=1,2,…,L, which are the vertices of the convex hull Γ.

The following hard constraints are taken into account so as to correspond with the engineering:

(3) {maxw⁡|[u(tk)]w|≤u¯,w∈{1,2,...,nu}maxj⁡|[x(tk)]j|≤x¯,j∈{1,2,...,nx}.

Communication protocols

In Fig. 1, the data are transmitted from sensors to controllers through a limited bandwidth shared communication network. In order to reduce the transmission burden, a hybrid scheduling strategy FRP is utilized to orchestrate the data transmission order. More specifically, only one sensor node has the right to access the shared network at each time instant, while others are transmitted according to corresponding strategy.

Figure 1 Structure of RMPC-based system via a high-rate network with the FRP.

The structure of FlexRay is shown in Fig. 2. It is composed of four parts, including static segment, dynamic segment, symbol window and network idle time (NIT). The part of static segment is composed of equal length time slots and transmits time-triggered periodic data. The deterministic time of message transmission can be ensured in the static segment, which is suitable for scenarios that need to transmit data at fixed time intervals. The dynamic segment uses flexible time division multiple access technology and can transmit event-driven signals that are transmitted only when needed, so it is able to support asynchronous processes. Thus, the flexible allocation of bandwidth can be achieved by the data volume requirement of periodic transmission and the event-driven amount in the network. In addition, the time slot in the dynamic segment is generally smaller than the time slot in the static segment, because the static segment needs enough time slots to ensure that the signal is transmitted successfully, while the signal in the dynamic segment is transmitted only when it is needed. The symbol window is a time slot to transmit data for network management purposes, when there is no symbol data to be transmitted, the symbol window can be configured to be of length 0, that is, no symbol data is sent. NIT is a period without communication so as to clock synchronization, it can provide a time buffer against possible communication delays or exceptions. The lengths of symbol window and NIT are quite small compared to the lengths of static and dynamic segments, therefore the time lengths of the former will be ignored.

Figure 2 Structure of FlexRay network.

In this article, the RRP and the TODP are implemented in the static and dynamic scheduling parts, respectively, and assume that there are N packets that should be sent from sensors to controller though the communication channel. The N packets are divided into two sets, where the first part h≥1 nodes belong to the set ℵ1=Δ{1,...,h} and the other nodes belong to the set ℵ2=Δ{h+1,...,N}. Taking varying real-time demands of sensor nodes into account and without loss of generality, in the set ℵ1 we use the RRP and the nodes belonging to set ℵ2 are scheduled by the TODP. Further elaboration regarding the process of identifying the chosen transmission nodes for the RRP and the TODP is provided in the following descriptions:

(1) RRP:

(4) ρ(i)=mod(i−1,h)+1

where ρ(i)∈ℵ1 is an element of ℵ1 and the ρth node has the right to access the communication channel at time i.

(2) TODP:

(5) ς(i)=arg⁡maxm=h+1,...N⁡{y~mT(i)χm(i)y~m(i)}

where “arg” refers to the minimum value of m that maximizing the quadratic, ς(i)∈ℵ2 denotes the node has the token at time instant i according to the rule of TODP, y~m(i)=Δym(i)−y¯m∗(i) represents the difference between the signal ym(i) at time i and the last transmitted signal y¯m∗(i) by node m before time i. χm(i)( m∈ℵ2) are the weighted matrices.

Remark1. The static segment of FRP uses time division multiple access technology, and the communication cycle is divided into dynamic segment and static segment time slots, and the frame ID is used to transmit the corresponding data. The static slot is fixed and is used to transmit periodic data. Each frame ID corresponds to a time slot and the communication node sends the message when the corresponding frame ID is assigned. The static slot frame ID is fixed and determined by the system designer in the initial stage, which is suitable for transmitting periodic and high reliability data. The smaller the frame ID of the dynamic segment, the higher the priority of the transmission data, which is suitable for the transmission of data that needs to obtain priority through competition, and the real-time requirement of the data is relatively low.

Suppose there are s measurement signals that are stacked into N packets and transmitted through the communication network. The FRP is utilized to orchestrate the order of the N network nodes. The measurement outputs of static segment and dynamic are given by

(6) {y[1]=Δ[y1T(⋅),y2T(⋅),....,yhT(⋅)]Ty[2]=Δ[yh+1T(⋅),yh+2T(⋅),....,yNT(⋅)]T

where yi(⋅)∈Rsi with Σi=1Nsi=s. The scheduling protocol of the first h measurement outputs are RRP while the remaining N−h outputs are scheduled by TODP.

Remark2. High-rate communication network is taken into account in this article so as to save communication resources and the order of sensors is determined by FRP. Before further discussion, the assumption is made for the high-rate communication network as follows:

Assumption1. The transmission signal communication period in the high-rate communication channels is defined as pc=p/(h+1) with pc as a constant and the first signal transmitted when the signal is sampled at 0th, that is to say t0=T0.

The measurement though the share communication network is defined as y¯(tk)=Δ[(y¯[1](tk))T,(y¯[2](tk))T]T, where y¯[1](tk)=Δ[y¯1T(tk),y¯2T(tk),...y¯hT(tk)]T and y¯[2](tk)=Δ[y¯h+1T(tk),y¯h+2T(tk),...y¯NT(tk)]T. y¯i(tk) i is the ith (i = 1,2,…N) element of the y¯(tk).

Then, in terms of Assumption 1, when the transmission occurs at time interval (tk−1,tk], the transmission instants can be expressed as tkp−hpc,tkp−(h−1)pc,...,tkp−pc,tkp, for the purpose of simplifying the expression, denote tki=Δtk−ipc(i=1,2,...,h). Next, how to combine the FRP with high-rate will be described. The RRP is used to schedule the transmission date at instant tkp−ipc(i=1,2,...,h) and the TODP is utilized to determine the node which most needed transmission at time tkp.

Next, according to the periodic feature of RRP and the characteristic of TODP, the zero-input strategy and the zero-order holder are adopted, respectively. The corresponding formulations are illustrated as follows:

(7) {y¯[1](tk)=[y1T(tkh)y2T(tkh−1)...yhT(tk1)]Ty¯[2](tk)=γς(tk)y[2](tk)+(IN2−γς(tk)y¯[2](tk−1)

where N2=ΔΣi=h+1Nsi, γς(tk)=Δ diag δ(h+1−ς(tk))Ish+1,...,δ(N−ς(tk))IsN. In more detail, it can be rewritten as

(8) {y¯[1](tk)=y[1](tk−1)y¯[2](tk)=γς(tk)y[2](tk)+(IN2−γς(tk)y¯[2](tk−1)=γς(tk)(c¯[2](tk)x(tk)+(IN2−γς(tk)y¯[2](tk−1)

where

c¯[1](tk)=Δ[c1T(tk)c2T(tk)c3T(tk)....chT(tk)]T,c¯[2](tk)=Δ[ch+1T(tk)ch+2T(tk)ch+3T(tk)....cNT(tk)]T.

According to Formula (8), y¯(tk) is defined as

(9) y¯(tk)=I1y[1](tk−1)+I2γς(tk)c¯[2](tk)x(tk)+I2(IN2−γς(tk)y¯[2](tk−1)

where I1=Δ[IN1×N1,0(s−N1)×N1T]T, N1=ΔΣi=1hsi, I2=Δ[0(s−N2)×N2T,IN2×N2]T.

Remark3. In this article, the assumption should highlight that the communication period pc is independent of the number of sensors. Because only one node has the transmission right when the communication channel transmits the signal in the high-rate network. In addition, without loss of generality, the communication period pc=p/(h+1) and h are prescribed in terms of engineering practice application.

Problem of interests

Generally speaking, the states of the practice system are difficult to obtain directly. Hence, the following controller with static output feedback strategy in the high-rate communication network is designed for the system Eq. (1) under the FRP.

(10) u(tk)=Fι(tk)(tk)y¯(tk),

ι(tk)={ϱ(tk),ι∈ℵ1ς(tk),ι∈ℵ2

where Fι(tk) is the feedback gain of the controller to be designed.

According to the control law Eq. (10), the form of closed-loop system can be rewritten as follows:

(11) η(tk+n+1|tk)=R¯ι(tk+n|tk)(tk)η(tk+n|tk),

where

(12) η(tk+n|tk)=[x(tk+n|tk)Ty[1](tk+n−1|tk)Ty¯[2](tk+n−1|tk)T]T,

R¯ι(tk+n|tk)(tk)=[Z(tk)B~(tk)I1B~(tk)I2γ¯ι(tk+n|tk)I1c¯[1](tk)00γι(tk+n|tk)c¯[2](tk)0γ¯ι(tk+n|tk)],

γ¯ι(tk+n|tk)=IN2−γι(tk+n|tk), Z(tk)=A(tk)+B(tk)Fι(tk+n|tk)γι(tk+n|tk)c¯[2](tk)I2,

B~(tk)=B(tk)Fι(tk+n|tk),

moreover, denote the initial state η(0)=[xT(0)(y[1](−1))T(y¯[2](−1))T]T.

The following “min-max” problem is put forward over infinite horizon for system Eq. (1) with polytopic uncertainties to design the controllers:

(13) minFι(tk+n|tk),⁡max(A(tk),B(tk),C(tk))∈Γ⁡J∞(tk),

where J∞(tk) is defined by

(14) J∞(tk)=Δ∑n=0∞⁡[ηT(tk+n)Qη(tk+n)+uT(tk+n)Ru(tk+n)]

where Q=diag{Q1,Q2,Q3} and R being prescribed symmetric and positive weighting matrices.

By virtue of the min-max problem Eq. (13), OP1 is illustrated to design the controllers:

OP1.minFι(tk+n|tk),⁡max(A(tk),B(tk),C(tk))∈Γ⁡J∞(tk),

s.t.maxw|[u(tk)]w|≤u¯,

s.t.maxj|[x(tk)]j|≤x¯,

η(tk+n|tk)∈φ(Pι(tk+n|tk),3τ),

where φ is terminal constraint set and is defined by

(15) φ=Δ{η(tk+n|tk)|ηT(tk+n|tk)Pι(tk+n|tk)η(tk+n|tk)≤3τ}

representing a quadratic function positive definite matrix with appropriate dimensions which can improve the performance of a system with polytopic uncertainties. It is noticeable that only the first component u(tk) of the predicted inputs set {u(tk),u(tk+1),u(tk+2),...} is valid for plant at each time instant.

In this article, we try to ensure the asymptotic stability of the system Eq. (1) in the framework of OFRMPC strategy and take both a high-rate network and FRP Eq. (8) into consideration. More precisely, in order to find the parameter matrices Fι(tk), an auxiliary optimization problem OP1 is put forward. Furthermore, the stability of the closed-loop system is ensured under the auxiliary optimization when the following requirements to all the admissible parameters are satisfied simultaneously: R1: A sub-optimization solution can be obtained from the auxiliary optimization problem which is utilized to represent the OP1.

R2: The closed-loop system is stable through the obtained parameter matrices Fι(tk).

Main results

MPC under FRP without hard constraints

In this section, some sufficient conditions are established to guarantee the performance of unconstrained systems by virtue of the quadratic function approach. Then, we will get the controllers in the framework of static OFRMPC strategy. First, sufficient conditions are given to satisfy the terminal constraint set condition in OP1, i.e., η(tk+n|tk)∈φ(Pι(tk+n|tk),3τ). Afterwards, an auxiliary optimization problem is put forward so as to seek the sub-optimal solution for the unconstrained systems. Furthermore, the unavailable state x(tk) of the established auxiliary optimization problem can be solved by inequality analysis technique. Then, considering the recursive solvability, another auxiliary optimization problem is constructed and the closed-loop system is stabilized by the sufficient conditions which are obtained from solving the online auxiliary optimization problem.

Terminal constraint set

The following definition with regard to robust positive invariant (RPI) is put forward.

Definition1. For a system, φ is a RPI set under the corresponding control law when the conditions η(tk)∈φ then η(tk+n)∈φ,n=0,1,2,... are satisfied.

On the basic optimization problem OP1, only if the following two conditions are satisfied simultaneously can the set φ(Pι(tk+n|tk),3τ) be guaranteed as the terminal constraint. C1: Select the quadratic function

(16) V(η(tk+n|tk))=ΔηT(tk+n|tk)Pι(tk+n|tk)η(tk+n|tk)

such that

(17) V(η(tk+n+1|tk))−V(η(tk+n|tk))≤−ηT(tk+n|tk)Qη(tk+n|tk)−uT(tk+n|tk)Ru(tk+n|tk).

C2: the set φ(Pι(tk+n|tk),3τ) is an RPI set.

Before further discussion, in order to simplify the expression, define ι(tk+n|tk)=c, ι(tk+n+1|tk)=z. The following theorem is provided to make sure the condition C1 is satisfied:

Lemma1. Give the positive-definite and symmetric matrices Q1,Q2,Q3 and R. For the system Eq. (11) controlled by Eq. (10), if there exists a positive scalar τ>0, symmetric and positively definite matrices Q~ic,Q~iz(i=1,2,3) and matrices Y2c,S^,S^c, for any (c,z) ∈S×S, l = 1, 2,…,L such that the following conditions hold:

(18) [Qcl→∗∗∗π~cτ¯I∗∗Y~c0τI∗Ξcl→00Q~z]≥0,

where

Qcl→=[Q¯1c000Q¯2c000Q¯3c],π~c=[Q1TlS000Q2S11000Q3S11],

Ξcl→=[AlTlS+BlγcS^cI2BlY2cBlγ¯cY2cI2I1S^00γcS^0γ¯cS11],

τ¯I=[τI000τI000τI],

Q~e=diag{Q~ie}(e∈{c,z},i=1,2,3),

Y~c=[RγcS^cI2RY2cRγ¯cY2c],

Q¯1c=(TlS)T+TlS−Q~1c,Q¯2c=(S11)T+S11−Q~2c,

Q¯3c=(S11)T+S11−Q~3c,Y2c=FcS11,S^=[S110],S^c=[Y2c0].

Then we can get the Eq. (17), furthermore, Pe=τQ~e−1(e∈{c,z}) and the feedback gain obtained by the control law Eq. (10) is as follows:

(19) Fc=Y2cS11−1

Proof: The token-dependent quadratic function is selected by Eq. (16), i.e.,

V(η(tk+n|tk))=ηT(tk+n|tk)Pcη(tk+n|tk),

where Pc=diag{Pic}(i=1,2,3) is the symmetric and positive-definite matrix to be designed.

The difference of Eq. (16) dependent on the system Eq. (11) is

(20) ΔV(η(tk+n|tk))=V(η(tk+n+1|tk))−V(η(tk+n|tk))=ηT(tk+n+1|tk)Pzη(tk+n+1|k)−ηT(tk+n|tk)Pcη(tk+n|tk)=ηT(tk+n|tk)(R¯cT(tk)PzR¯c(tk)−Pc)η(tk+n|tk).

Then, define Tl=[(c¯l)T(c¯l(c¯l)T)−1(c¯l)⊥], where c¯l=[c¯[1]lc¯[2]l],(c¯l)⊥ denotes the orthogonal basis of the null space for c¯l and introduces the following free matrix:

(21) S=[S1100S22]

where S11 and S22 are the arbitrary matrix with appropriate dimensions.

The following conditions:

TlS+(TlS)T−Q~1c−(TlS)TQ~1c−1TlS=−(Q~1c−TlS)Q~1c−1(Q~1c−TlS)T≤0,S11+(S11)T−Q~ic−(S11)TQ~ic−1S11=−(Q~ic−S11)Q~ic−1(Q~ic−S11)T≤0,(i=2,3),

are substituted into Eq. (18) and we will obtain

(22) [Qcl←∗∗∗π~cτ¯I∗∗Y~c0τI∗Ξcl→00Q~z]≥0,

where

Qcl←=[(TlS)TQ~1c−1TlS000S11TQ~2c−1S11000(S11)TQ~3c−1S11].

After that, it should be noticed that if we pre-and post-multiplying the following inequalities with diag {TlS,I,S11,I,...,I⏟7} and its transpose, we will obtain Eq. (22).

(23) [Q→c∗∗∗π¯cτ¯I∗∗Y¯c0τI∗Ξcl←00Qz~]≥0,

where

Q→c=[Q~1c−1000S11TQ~2c−1S11000Q~3c−1],π¯=[Q1000S11Q2000Q3],

Ξcl←=[Al+BlFcγcc¯[2]lI2BlY2cI1BlFcI2γ¯cI1c¯[1]l00γcc¯[2]l0γ¯c],

Y¯c=[RFcγcc¯[2]lI2RY2cRFcγ¯c].

Then, the following inequalities Eq. (24) can be easily obtained by pre-and post-multiplying Eq. (23) with diag {I,S11−1,I,...,I⏟8} and its transpose.

(24) [Q~c−1∗∗∗Qτ¯∗∗Ycˇ0τI∗R¯cl00Q~z]≥0,

where Q=diag{Q1,Q2,Q3},

Ycˇ=[RFcγcc¯[2]lI2RFcI1RFcγ¯c,]

(25) R¯cl=[Al+BlFcγcc¯[2]lI2BlFcI1BlFcI2γ¯cI1c¯[1]l00γcc¯[2]l0γ¯c].

Because the system Eq. (11) is a polytopic uncertainties system, this implies Eq. (24) is affine in Γ, and the elements of convex hull have

(26) [Q~c−1∗∗∗Qτ¯∗∗Ycˇ0τI∗R¯c(tk)00Q~z]≥0.

By virtue of the Schur complement, it is easily obtained from Eq. (26) that

(27) R¯cT(tk)Q~zR¯c(tk)−Q~c+τ−1Q+τ−1YcˇTYcˇ≤0.

Let Eq. (27) be multiplied by τ>0 and define the matrix pe=τQ~e−1(e∈{c,z}) so that we can obtain Eq. (28).

(28) R¯cT(tk)Q~zR¯c(tk)−Pc+Q+YcˇTYcˇ≤0

pre-and post-multiplying Eq. (28) with ηT(tk+n|tk) and its transpose

ωc(tk)−ηT(tk+n|tk)Pcη(tk+n|tk)+ηT(tk+n|tk)Qη(tk+n|tk)+ℓc≤0.

where

ωc=(R¯c(tk)η(tk+n|tk))TPzR¯c(tk)η(tk+n|tk).

ℓc=(Ycˇη(tk+n|tk))TYcˇη(tk+n|tk).

It is obvious that condition Eq. (17) i.e., condition C1, can be guaranteed by Eq. (28). Then, the proof is complete.

Remark4. In spite of the application of FRP in the high-rate communication channel alleviating the data conflict and network congestion, there are myriad side-effects that might be produced because the information transmission order is changed. For the sake of dealing with the problem, we employ the token-dependent Lyapunov-like approach. In this way, the affection of FRP in the high-rate communication channel can be well reflected. Moreover, the proposed analysis method can better reflect the scheduling characteristics of the communication protocol compared with traditional Lyapunov function. Zou, Wang & Gao (2016) introduced a technical method on how to choose different token-dependent Lyapunov functions based on various communication protocols.

Then, we pay attention to the condition C2 which is needed to find the sufficient conditions such that the set φ(Pι(tk+n|tk),3τ) is an RPI set. The way is illustrated as follows:

On the basic of definition 1, both of the following requirements must be satisfied simultaneously to ensure the set φ(Pι(tk+n|tk),3τ) is an RPI: (∅1) the set φ contains the initial state of the system Eq. (1), i.e.,

ηT(tk)Qc−1η(tk)≤3;

(∅2) all the future states η(tk+n|tk),n=1,2,... also in the set φ.

Next, we will deal with the above requirements one by one.

Through the technique of Schur complement, (∅1) holds if

(29) [3∗η(tk)Qc]≥0.

After that, in terms of Lemma 1 and Eq. (29), it can be easily inferred that

(30) V(η(tk+n+1))≤V(η(tk+n))≤...≤V(η(tk))≤3τ

which indicates, as long as the initial state η(tk) of the system belongs to the set φ, all the future states η(tk+n|tk) pertain to the set φ. The proof of RPI is then completed.

Until now, conditions Eqs. (18) and (29) can ensure the terminal constraint set. That is to say, the condition η(tk+n|tk)∈φ(Pc,3τ) is satisfied.

Auxiliary optimization problems

In this section, the unconstrained system with polytopic uncertainties for OP1 is discussed.

It should be noted that OP1 is an infinite horizon optimization problem that involves parameter uncertainties, which make it extremely challenging to deal with directly. Thus, an auxiliary optimization problem is introduced as an alternative approach to find the optimal solution. Next, we will try to present the aforementioned auxiliary problem.

It is easily seen that η(∞|tk)=0 and V(∞)=0 from Eq. (17). By summing both sides of Eq. (17) from n = 0 to n = ∞ and utilizing Eq. (14), we obtain

(31) J∞(tk)≤V(tk)=ηT(tk)Pcη(tk)≤3τ,

which indicates

(32) max(A(tk),B(tk),C(tk))∈Γ⁡J∞(tk)≤3τ.

Then, the min-max optimization problem of OP1 is transformed into an minimization auxiliary problem OP2, which finds the upper bound of object function. It is described as follows:

OP2.minQ~ic>0.Q~iz>0(i=1,2,3),(c,z)∈S×S,Fc⁡3τ,s.t. Eqs. (18) and (29).

It is worth mentioning that not all the states are measurable, therefore, the problem of unavailable states which influence the condition Eq. (29) verification online should be solved. Before that, the following assumption is proposed for the Lemma 2 to deal with the obstacle.

Assumption2. The initial state of the system Eq. (1) belongs to the following set:

(33) x(0)∈{x(tk)|x(tk)TS−1x(tk)≤1},

where S>0 is a matrix predetermined by a practical engineering project.

Lemma2. Given Assumption 2 and if there exists symmetric and positive matrices Qbcˇ(b=1,2,3) for system m Eq. (1) controlled by Eq. (10), such that

(34) [2∗∗y[1](tk−1)Q~2c∗y¯[2](tk−1)0Q~3c]≥0.

(35) [Q~c∗R¯clQ~cQcˇ]≥0.

(36) Q~1c≥Q1cˇ,Q~1c≥S,Q~2c≥Q2cˇ,Q~3c≥Q3cˇ

(37) [Q~c∗∗[0,I,0]R¯clQ~c2/3Q2cˇ∗[0,0,I]R¯clQ~c02/3Q3cˇ]≥0.

holds, where Qcˇ=Δ diag {Q1cˇ,Q2cˇ,Q3cˇ}, the vertices of R¯c(tk),(l=1,2,3,...,L) is represented by R¯cl, after that, the condition Eq. (29) can always be confirmed.

Proof: By virtue of the method similar to Song, Wei & Liu (2016), the above lemma can be obtained. Thus, the proof is omitted here.

Through the technique of Schur complement and the Lemma 1, the following inequality can be obtained from Eqs. (35) and (37), respectively,

(38) [Qcl→∗Ξcl→Qcˇ]≥0.

(39) [Qcl→∗∗Jc2/3Q2cˇ∗Fc02/3Q3cˇ]≥0.

where

Jc=[0Q~2c0],

Fc=[γcS^0γ¯cS11].

Considering both Assumption 2 and Lemma 2, the optimization problem OP2 is transformed into the following problem which finds the approximate optimization solution of unmeasurable states:

OP3.minQ~ic>0,Q~iz>0,Qicˇ>0(i=1,2,3),(c,z)∈S×S,Fc⁡3τ,

s.t. Eqs. (18), (33), (34) and (36), (38), (39).

Feasibility and stability

In this part, the feasibility of the proposed problem and the stability of the system Eq. (1) are demonstrated.

Theorem1. Let us consider the symmetric and positive-definite matrices Q1, Q2,Q3 and R be given. Furthermore, the system Eq. (1) with polotypic uncertainties is controlled by Eq. (10). After that, if there exists a feasible solution to OP3 at initial time tk, then it remains feasible at any future time instant tk+n>tk,(n=1,2...). Moreover, the system is stable as well as the feedback gains that can be acquired by Eq. (19).

Proof: (1) Feasibility. It is easily seen that only the condition Eq. (18) is related to the states, for this reason we only need to demonstrate the feasibility of condition Eq. (18) in the prediction horizon, that is to say, it merely needs to guarantee the feasibility of condition Eq. (29).

It can be obtained from systems Eq. (1) and Eq. (11) that

(40) η(tk+n|tk+n)=η(tk+n),n≥1

(41) η(tk+1|tk)=R¯ι(tk)(tk)η(tk|tk)

(42) η(tk+1)=R¯ι(tk)(tk)η(tk).

On the basic of the characteristic of RPI and R¯ι(tk)(tk)∈Γ, it is easily seen that

(43) ηT(tk+1|tk)Q~z−1η(tk+1|tk)<ηT(tk)Q~c−1η(tk)<3.

The following condition can be inferred by virtue of Eqs. (41–43).

(44) ηT(tk+1)Q~z−1η(tk+1)<3.

This indicates the feasibility of condition Eq. (29) is satisfied at time tk+1 and we can get the same result for all the future times tk+2,tk+3,... through the recursion method.

(2) Stability. The asymptotic stability of the system Eq. (1) controlled by Eq. (10) can be proved by establishing the decreasing quadratic function Vˇ(η(tk))=ηT(tk)P∗ι(tk)η(tk), where the notation “∗” denotes the optimal solution of OP3. The following can be obtained from the proof of feasibility.

(45) ηT(tk+1)Pι(tk+1)η(tk+1)≤ηT(tk)Pι(tk)η(tk)<ηT(tk)Pι(tk)∗η(tk),

where Pi(i∈{ι(tk),ι(tk+1)}) represents the feasible solution. Then, we have

(46) ηT(tk+1)Pι(tk+1)∗η(tk+1)≤ηT(tk+1)Pι(tk+1)η(tk+1),

through the characteristic of optimal solutions and feasible solutions. Combining Eqs. (45) and (46) yields

(47) ηT(tk+1)Pι(tk+1)∗η(tk+1)≤ηT(tk)Pι(tk)∗η(tk),

hence, the quadratic function V~(η(tk)) decreases strictly and the proof of theorem is complete.

MPC under FRP with hard constraints

On the basis of conditions currently established, this section focuses on addressing the MPC problem for polotypic systems subject to hard constraints under the FRP. Afterwards, myriad sufficient conditions are acquired. Lastly, an algorithm is introduced to deal with an online optimization problem subject to certain conditions.

Controller design under FRP

First, some inequalities are provided while the hard constraints of inputs and states Eq. (3) are taken into account. After that, the controllers are designed for a constrained system in the framework of RMPC by the optimization problem and propose the corresponding algorithm.

Lemma3. If there exists symmetric and positive definite matrices S11,Q~1c,Y2c and the following conditions for any (c,z)∈R×R is satisfied. Then, the hard constraints Eq. (3) are met simultaneously.

(48) [I∗Y2cTu¯2s11]≥0.

(49) [I∗Q~1cx¯2Q~1c]≥0.

Proof: According to the Cauchy-Schwarz inequality technique, Eq. (19) can be converted to

(50) |[u(tk+n|tk)]w|2=|vw(Fcy¯(tk+n|tk))|2=|vwY2cs11−1y¯(tk+n|tk)|2=|vwY2cs11−1/2s11−1/2y¯(tk+n|tk)|2≤∥vwY2cs11−1/2∥2∥s11−1/2y¯(tk+n|tk)∥2≤∥vwY2cs11−1/2∥2=vw(Y2cs11−1Y2cT)vwT≤u¯2

where vw represents the wth row of an nu-ordered identity matrix. Equation (50) is guaranteed by Eq. (48), which implies the input constraints are satisfied. Then, the constraint on the state predictions can be obtained in light of the same way shown above, and get

(51) |[x(tk+n|k)]j|2=|vjQ~1c1/2Q~1c−1/2x(tk+n|k)|2≤∥vjQ~1c1/2∥2∥Q1c−1/2x(tk+n|k)∥2≤∥vjQ~1c1/2∥2=vjQ~2cvjT≤x¯2

where vj is the jth row of an nx-ordered identity matrix.

In terms of Lemma 3, for the system Eq. (1) with hard constraint under FRP in the high-rate network, the further auxiliary optimization problem is illustrated as

OP4.minQ~ic>0,Q~iz>0,Qicˇ>0(i=1,2,3),(c,z)∈S×S,Fc⁡3τ,

s.t. Eqs. (18), (33), (34) and (36), (38), (39) and (48), (49).

On the basis of previous discussions, we prepare to show the following theorem so that the constraints system Eq. (1) under FRP controlled by Eq. (10) is asymptotically stable.

Theorem2. The system Eq. (1) with hard constraints and the FRP in the high-rate communication channel are taken into account, Eqs. (1) and (3) controlled by Eq. (10). Assume that the OP4 is feasible at the initial time tk, then the feasibility for any future time instant tk+n>tk,n>1 is satisfied. Moreover, system is stable by employing the feedback gains Fc=Y2cs11−1.

Proof: The proof process is omitted here since it is similar to Theorem 1.

Algorithm of MPC for constrained system under FRP

In this section, taking into consideration system Eq. (1) with hard constraints in the framework of static OFMPC strategy over the high-rate network under FRP, an algorithm is put forward.

Algorithm: Off−linepart:	
Select the initial state η(0)=[xT(0)y[1](0)y¯[2](0)] and matrix S with appropriate dimensions such that x(0)∈{x(tk)|x(tk)TS−1x(tk)≤1} is feasible at initial time instant.	
On−linepart:	
Step1. Through the parameters from the Off−linepart, the controller gain Fc is obtained by solving the OP4.	
Step2. Calculate Fc=Y2cs11−1 and act u(tk)=Fcy¯(tk) on the plant and go back to Step1.	

Remark5. For discrete time linear systems with polytopic uncertainties under FRP in the high-rate communication channel, the RMPC problem is handled. It is noticeable that the distinction of our results can be illustrated as follows: (1) due to the limited communication capacity of the network channels existing constraint, for systems with polytopic uncertainties, the FRP combined with a high-rate network is introduced to handle the static OFRMPC issue; (2) because of the coupling between the high-rate communication channels and the FRP, a periodic sequence is employed to model the complex transmission mechanism of the sensors; (3) a few token-dependent methodologies and efforts are made to obtain the desired results under the FRP; (4) the optimization problem OP4 is presented to find a certain upper bound of objective function; (5) an online static OFRMPC algorithm is put forward to get myriad controllers, which make the closed-loop system with hard constraint asymptotically stable.

Illustrative example

In this section, the effectiveness of the proposed OFRMPC strategy is validated via two examples.

A. Example 1:

A distillation process (Al-Gherwi, Budman & Elkamel, 2011) has the reflux and boil-up manipulated variables, the controlled variables include top and bottom composition. From the practical view point, we take parameter uncertainties into consideration and select the same period as Al-Gherwi, Budman & Elkamel (2011) to obtain the following discrete-system model.

(52) x(tk+1)=[0.9481000.9481]x(tk)+(−20)∗[0.5120.01680.09760.539]u(tk)=ΔAx(tk)+Bu(tk),

y(tk)=[1.5001.5]x(tk)=ΔCx(tk).

Select the initial value x(0)=[−5,5]T, y(−1)=[0.5,0.5]T, y¯[2](−1)=[0.8,0.8]T. The constraints of input and state are taken as u¯=650 and x¯=950, respectively. The weighting matrix S=[0.95000.95], furthermore the parameters of system are given as follows:

A(1)=A(2)=A=[0.9481000.9481],

C(1)=C(2)=[1.5001.5],

Q1=Q2=Q3=[3003],R=0.01∗[0.05000.05].

Figure 3 depicts the evolution of inputs and states of the closed-loop system, which confirm the system subject to hard constraints over a high-rate network under FRP is stable by virtue of the proposed static OFRMPC strategy. Figure 4 plots the state responses for the open-loop and closed-loop systems. It is obvious that the systems converge to stability faster under the designed control law. Figure 5 compares RRP and shows that FRP can make the system state have faster convergence speed. Figure 6 is the comparison diagram of the system state for three different number of cycles in the high-rate network, which shows that the more the number of cycles, the faster the convergence speed of the system state.

Figure 3 The evolution of states and inputs of the closed-loop system.

Figure 4 Comparison of state evolution of open-loop and closed-loop systems.

Figure 5 State evolution of FRP and RRP.

Figure 6 Closed-loop system state trend under different communication cycles.

B. Example 2:

The second example is presented without any control and the parameters are provided as follows:

A(1)=A=[10.1−0.13−0.1],A(2)=[1.10.15−0.1−0.2],

B(1)=B(2)=[25.016.632.4221.94],

C(1)=C(2)=[1.5001.5].

Figure 7 plots the trajectories of the states and the control input of the closed-loop system and we can easily see that the system can be stable under the designed controller. The state comparisons between the open-loop system and closed-loop system are depicted in Fig. 8. It can be found that the system without control can not converge. Figure 9 clearly shows that FRP can make the system reach stability faster. Figure 10 illustrates that the system can be more stable as M increases.

Figure 7 The evolution of states and inputs of the closed-loop system.

Figure 8 State responses for the open-loop and closed-loop systems.

Figure 9 Comparison of state convergence speed between FRP and RRP.

Figure 10 System state changes of FRP under different communication cycles.

Conclusions

In this article, the static OFRMPC problem for the uncertain system over a high-rate network under FRP is investigated. To reduce data collision as well as improve flexibility, the FRP which includes static segment and dynamic segment is utilized during signal transmission, e.g., only one node has the right to access the share communication channel at each time instant, the nodes of static segment adopted zero-input strategy and the others employed zero-order holder ZOH. Moreover, the high-rate network has been introduced so as to enhance the utilization of communication resources. Sufficient conditions have been obtained to guarantee the stability of controllers in the framework of OFRMPC with the support of Lyapunov stability theory. Furthermore, the parameters of controllers have been obtained by dealing with matrix inequalities. Finally, two simulation examples have been utilized to validate the effectiveness of the proposed algorithm.

Supplemental Information

Supplemental Information 1 Numerical simulation 1 is given to validate the effectiveness of the proposed controller design scheme.

A distillation process has the reflux and boil-up manipulated variables, the controlled variables include top and bottom composition. From the practical view point, we take parameter uncertainties into consideration and select the period to obtain the following discrete-system model.

Supplemental Information 2 Numerical simulation 2 is given to validate the effectiveness of the proposed controller design scheme.

The second example (the original system is divergent) is presented to validate the effectiveness of MPC controller.

Additional Information and Declarations

Competing Interests

Author Contributions

Data Availability

The authors declare that they have no competing interests.

Jianhua Wang conceived and designed the experiments, performed the experiments, analyzed the data, performed the computation work, prepared figures and/or tables, authored or reviewed drafts of the article, and approved the final draft.

Fuqiang Fan conceived and designed the experiments, performed the experiments, analyzed the data, performed the computation work, prepared figures and/or tables, authored or reviewed drafts of the article, and approved the final draft.

Yanye Yu conceived and designed the experiments, performed the experiments, analyzed the data, performed the computation work, prepared figures and/or tables, authored or reviewed drafts of the article, and approved the final draft.

Shuxin Du conceived and designed the experiments, performed the experiments, analyzed the data, performed the computation work, prepared figures and/or tables, authored or reviewed drafts of the article, and approved the final draft.

Xiaorui Guo conceived and designed the experiments, prepared figures and/or tables, authored or reviewed drafts of the article, searched the database, analyzed database search results, and approved the final draft.

The following information was supplied regarding data availability:

The two simulation examples for demonstrating the effectiveness of the proposed MPC algorithm are available in the Supplemental Files.

The system parameters of Example 1 are available in this article: doi:10.1016/j.jprocont.2011.07.002.

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
