# Peer review of "Robust model predictive control for polytopic uncertain systems via a high-rate network with the FlexRay protocol"

_PeerJ Computer Science, doi:10.7717/peerj-cs.2580_

## Round 0.1 · original submission · Major Revisions

After the first round of review, a "Major Revision" is recommended for this paper. Authors should carefully review all the comments, in particular to clearly explain the proposed network protocol FlexRay and enhance the validation of the proposed method.

Reviewer 1 ·

Basic reporting

no comment

Experimental design

no comment

Validity of the findings

no comment

Additional comments

no comment

Annotated reviews are not available for download in order to protect the identity of reviewers who chose to remain anonymous.

·

Basic reporting

The layout of Figure 1 is unreasonable. It is recommended to modify it to increase readability.

Experimental design

The name of "Control Inpuut" in Figure 3 and Figure 5 is wrong, please correct it.

Validity of the findings

In the experimental simulation part of this article, only using two examples for verification is not enough to prove the effectiveness. It is recommended to use 3-5 simulation examples for verification.

Additional comments

FRY is usually used in vehicle communications. Can the RMPC problem proposed in this article be applied to actual cars? Can it be simulated using CANoe and other software, or tested in actual vehicles.

Reviewer 3 ·

Basic reporting

it meets the requirements.

Experimental design

Contents in the manuscript pertaining to control theory. Manuscript contains control aspect more than the network mentioned in the title.

Validity of the findings

Illustrated with some examples.

Additional comments

Notations used should be clearly defined before its use. Some of the variables are described after its use.

Annotated reviews are not available for download in order to protect the identity of reviewers who chose to remain anonymous.

Reviewer 4 ·

Basic reporting

The paper manuscript is well structured, communicating the research topic and contributions. In the introduction part, the author provides a solid background, demonstrating the values of Model Predictive Control (MPC) and the challenges associated with polytopic uncertain systems. The literature is well-referenced.

The paper adheres to standard scientific reporting norms, with each section logically progressing from the introduction to the experimental results and conclusions. Figures and tables are pertinent and well-integrated.

Experimental design

The experimental design is well-suited to the research question, addressing the challenge of implementing RMPC in high-rate networks governed by the FlexRay protocol. The research question is clearly defined and significant, tackling the complexities of controlling systems with uncertain parameters in a networked environment.

The methodology is sufficiently detailed to enable replication, with rigorous presentation of the mathematical formulations and RMPC strategy development. However, the paper could be strengthened by offering more detailed explanations of the algorithms used, particularly the rationale behind the selection of specific mathematical models and protocols. F

In Line 70: for example, why is a hybrid of both event-triggering and time-triggering superior than a single case?

Validity of the findings

The findings are valid and well-supported by the data. The authors effectively demonstrate that the proposed RMPC strategy manages system uncertainties while maintaining stability and performance. The simulations are compelling, illustrating that the proposed method surpasses traditional approaches in managing high-rate network constraints.

Nonetheless, the paper would benefit from a deeper analysis of the proposed method's limitations.

For example, discussing the potential effects of network delays or packet loss on RMPC performance would provide a more comprehensive understanding of the system’s robustness.

Additional comments

Conclusions: The conclusions are well-grounded, summarizing the key findings and their implications for future research in networked control systems. The authors successfully argue that their proposed RMPC strategy offers a viable solution for managing uncertainties in polytopic systems over high-rate networks. However, the paper could be improved by discussing the practical implementation of the proposed strategy in real-world scenarios where network conditions may vary.


General Comments: This paper contributes to the field of robust control systems, particularly within the context of high-rate networked environments. The integration of the FlexRay protocol with RMPC is novel and addresses a clear gap in the literature. The manuscript is well-structured, with a logical flow and adequate detail to support the authors’ claims.

While the paper is well-structured, numerous basic grammatical errors may hinder the reader's ability to understand the content seamlessly. I have highlighted a few examples below, but there are far more such errors! The authors are strongly encouraged to use proofreading tools to address all grammatical issues before publication.

Line 29: 'etcZhou?' (requires correction)
Line 29: 'It should be point out' -> 'It should be pointed out'
Line 30: 'Since' should be followed by a clause rather than the noun 'its capacity'
Line 35: 'a rich body results' -> 'a rich body of results'
Line 40: 'some research get the results' -> 'some research gets the results'
Line 41: 'could well ensured' -> 'could be well ensured'
Line 42: 'due to' should be followed by a noun, not a clause.
Line 46: 'because its advantages' -> 'because of its advantages'
Line 50: 'however, it exist the shortage such as' -> 'However, some shortages still exist, such as'
Line 70: Duplicate use of the word 'hybrid'"

---

## Round 0.2 · Minor Revisions

Authors have addressed most of the comments from the reviewers satisfactory. However, there are still some concerns about lacking of the explanation of the operation of FlexRay protocol and also clarification of whether updating of feedback gain is not necessary required in every sampling time. Authors need to carefully address these comments.

·

Basic reporting

The content of the paper basically meets the requirements of the revision suggestions.

Experimental design

The experimental method is reasonable and the data analysis is quite reasonable.

Validity of the findings

No comment

Reviewer 3 ·

Basic reporting

This article comes under the area of networked control systems

Experimental design

FlexRay protocol description may be added

Validity of the findings

No comments

Additional comments

The author's of this manuscript have made revisions based on the reviewer's comments and improved the quality of the paper. However, the authors can further modify it to improve the quality of the paper by adding the working of FlexRay protocol. In addition, authors need to clarify, whether updating of feedback gain is not necessary required in every sampling time.

Reviewer 4 ·

Basic reporting

The revised paper and response letter address all my comments from the previous review round. The paper follows established scientific reporting conventions, with each section flowing logically from the introduction to the experimental results and conclusions.

Experimental design

1. Remark 1 on page 5 of the revised paper is extended which addresses my question about the the advantages of both event-triggering and time-triggering over a single case.

2. The concern of "Robustness of the system" is well explained and answered.

Validity of the findings

The findings are valid and well-supported by the data.

Additional comments

All the grammatical issues are resolved which enhances the reliability of the paper.

---

## Round 0.3 · accepted · Accept

Authors have addressed all the comments from the reviewers. Hence, this paper is recommended to be accepted in its current form.

Reviewer 1 ·

Basic reporting

No comment.

Experimental design

No comments.

Validity of the findings

No comments.

Additional comments

No.

Annotated reviews are not available for download in order to protect the identity of reviewers who chose to remain anonymous.

·

Basic reporting

The structure of the article is basically reasonable. According to the suggested modifications, the content has been comprehensively revised.

Experimental design

The experimental analysis is relatively clear.

Validity of the findings

No comment.

Reviewer 3 ·

Basic reporting

The revised manuscript has significantly improved with respect to the original version and have made revision based on the reviewer's comment.

Experimental design

No comment

Validity of the findings

No comment

Additional comments

No comment